# TGF-β Regulates Collagen Type I Expression in Myoblasts and Myotubes via Transient *Ctgf* and *Fgf-2* Expression

**DOI:** 10.3390/cells9020375

**Published:** 2020-02-06

**Authors:** Michèle M. G. Hillege, Ricardo A. Galli Caro, Carla Offringa, Gerard M. J. de Wit, Richard T. Jaspers, Willem M. H. Hoogaars

**Affiliations:** Laboratory for Myology, Department of Human Movement Sciences, Faculty of Behavioural and Movement Sciences, Vrije Universiteit Amsterdam, Amsterdam Movement Sciences, 1081 BT Amsterdam, The Netherlands; m.m.g.hillege@vu.nl (M.M.G.H.); r.a.galli@amsterdamumc.nl (R.A.G.C.); c.offringa@vu.nl (C.O.); g.m.j.de.wit@vu.nl (G.M.J.d.W.); w.m.h.hoogaars@umcg.nl (W.M.H.H.)

**Keywords:** *Acvr1b*, *Tgfbr1*, myostatin, *Col1a1*, skeletal muscle, fibrosis, myogenesis, atrophy

## Abstract

Transforming Growth Factor β (TGF-β) is involved in fibrosis as well as the regulation of muscle mass, and contributes to the progressive pathology of muscle wasting disorders. However, little is known regarding the time-dependent signalling of TGF-β in myoblasts and myotubes, as well as how TGF-β affects collagen type I expression and the phenotypes of these cells. Here, we assessed effects of TGF-β on gene expression in C2C12 myoblasts and myotubes after 1, 3, 9, 24 and 48 h treatment. In myoblasts, various myogenic genes were repressed after 9, 24 and 48 h, while in myotubes only a reduction in *Myh3* expression was observed. In both myoblasts and myotubes, TGF-β acutely induced the expression of a subset of genes involved in fibrosis, such as *Ctgf* and *Fgf-2,* which was subsequently followed by increased expression of *Col1a1*. Knockdown of *Ctgf* and *Fgf-2* resulted in a lower *Col1a1* expression level. Furthermore, the effects of TGF-β on myogenic and fibrotic gene expression were more pronounced than those of myostatin, and knockdown of TGF-β type I receptor *Tgfbr1*, but not receptor *Acvr1b,* resulted in a reduction in *Ctgf* and *Col1a1* expression. These results indicate that, during muscle regeneration, TGF-β induces fibrosis via *Tgfbr1* by stimulating the autocrine signalling of *Ctgf* and *Fgf-2.*

## 1. Introduction

Muscle wasting disorders, such as sarcopenia, cachexia and muscle dystrophies, are characterised by muscle fibre injury or atrophy, which results in the gradual replacement of muscle fibres by adipose and fibrotic tissue [1,2]. This leads to progressive muscle weakness and loss of contractile function. Transforming Growth Factor β (TGF-β) is known for its role in the regulation of skeletal muscle size as well as fibrosis and contributes to the progressive pathology of muscle wasting disorders such as Duchenne Muscular Dystrophy (DMD) [3,4]. 

TGF-β functions by regulating expression of target genes via specific binding of type II and type I receptor kinases and subsequent activation of intracellular receptor-regulated SMAD2 and SMAD3 proteins (R-SMADS) [5]. TGF-β is expressed by multiple cell types, such as macrophages, monocytes, neutrophils, fibroblasts and bone cells [6,7,8,9]. While TGF-β is transiently expressed during skeletal muscle regeneration following injury [10], prolonged elevated TGF-β protein levels are associated with pathologies such as DMD [3], limb girdle muscular dystrophy and amyotrophic lateral sclerosis (ALS), as well as sarcopenia [11,12,13]. TGF-β may affect skeletal muscle size by the inhibition of muscle stem cell (MuSC) differentiation and the induction of the atrophy of muscle fibres. In vitro studies have shown that TGF-β inhibits myoblast differentiation through the repression of myogenic gene expression, whereas differentiated myotubes seem to be insensitive to TGF-β-induced myogenic inhibition [14,15,16]. Muscle-specific overexpression of TGF-β in mice stimulates the expression of E3 ligase (i.e., atrogin-1) and concomitant muscle atrophy [17,18]. However, whether the induction of atrogin-1 and muscle atrophy is a direct effect of TGF-β expression or an indirect effect via the stimulation of other paracrine factors remains to be assessed. 

TGF-β is also known to be involved in fibrosis. Overexpression of TGF-β in mouse skeletal muscle results in excessive collagen deposition [17]. In addition, antibody treatment to neutralise TGF-β in murine X-linked muscular dystrophy (mdx) mice reduces connective tissue deposition compared to that of untreated mdx mice [19]. Moreover, C2C12 myoblasts overexpressing TGF-β transdifferentiate into fibrotic cells after transplantation into skeletal muscle, which indicates that muscle cells may contribute to fibrosis [20].

Another TGF-β family member, muscle specific cytokine myostatin, has been shown to inhibit myoblast differentiation via a similar mechanism as via TGF-β [21]. Furthermore, myostatin is a well-known regulator of muscle mass and has been suggested to be involved in muscle fibrosis [22]. Myostatin signals via distinct type II and type I receptors than TGF-β does, but also through phosphorylation of SMAD2/3 [23,24]. TGF-β signals mainly via the type I receptor TGF-β receptor type-I (TGFR-1) [24]. While in muscle cells myostatin signals mainly via type I receptor Activin receptor type-1B (ACTR-1B)*,* in fibroblasts myostatin signals mainly via TGFR-1 [23,25]. Both proteins have been indicated as possible therapeutic targets for muscle wasting disorders.

While transient TGF-β expression may contribute to muscle regeneration after injury, the chronic elevated expression of TGF-β in skeletal muscle may be detrimental [cf.10]. Although the role of TGF-β in muscle mass regulation and skeletal muscle fibrosis has been studied extensively, the effects on myoblasts and differentiated muscle cells and underlying mechanisms are not well understood. The aim of this study was to assess the time-dependent effects of TGF-β signalling and downstream signalling on the expression of myogenic, atrophic and fibrotic genes in both myoblasts and myotubes. Furthermore, taking into account the functional and mechanistic similarities between TGF-β and myostatin, as well as the fact that both ligands have been implied as possible therapeutic targets for muscle wasting disorders, the effects of TGF-β and myostatin signalling in myoblasts were compared. Our data indicate that TGF-β inhibits myogenic gene expression in both myoblasts and myotubes but does not affect myotube size. Most importantly, our results show that TGF-β stimulates collagen type I, alpha 1 (*Col1a1*) mRNA expression in both myoblasts and myotubes, which is largely induced via autocrine expression of connective tissue growth factor (*Ctgf*) and fibroblast growth factor-2 (*Fgf-2*). Lastly, the effects of TGF-β on myogenic and fibrotic signalling are more pronounced than those of myostatin, and only TGF-β receptor type-I (*Tgfbr1*) mRNA knockdown, but not Activin receptor type-1B (*Acvr1b*) mRNA knockdown, decreased *Ctgf* and *Col1a1* expression levels, suggesting that myoblasts are more sensitive to TGF-β than to myostatin.

## 2. Materials and Methods

### 2.1. C2C12 Cell Culture

The C2C12 mouse muscle myoblast cell line (ATCC CRL-1772) was obtained from ATCC (Wesel, Germany). Cells were cultured in growth medium (DMEM, 4.5% glucose (Gibco, 11995, Waltham, MA, USA), containing 10% fetal bovine serum (Biowest, S181B, Nuaillé, France), 1% penicillin/streptomycin (Gibco, 15140, Waltham, MA, USA), and 0.5% amphotericin B (Gibco, 15290-026, Waltham, MA, USA)) at 37 °C, 5% CO_2_. The cells were used for experiments between passage 4–14. All experiments with C2C12 cells were performed on collagen-coated plates (collagen I rat protein, tail (Gibco, A10483-01, Waltham, MA, USA) diluted in 0.02N acetic acid). C2C12 myoblasts were cultured in differentiation medium (DMEM, 4.5% glucose, 2% horse serum (HyClone, 10407223, Marlborough, MA, USA), 1% penicillin/streptomycin, 0.5% Amphotericin B) for 16 h or allowed to differentiate for 3 days before treatment. Cells were treated with 10 ng/mL TGF-β1 (Peprotech, 100-21C, London, UK) or 300 ng/mL myostatin (Peprotech, 120-00, London, UK) for 0, 1, 3, 9, 24 or 48 h, unless indicated differently. The cells were treated with 10µM Ly364947 (dissolved in dimethyl sulfoxide (DMSO), 1mM). As a control, cells were treated with 0.1% DMSO.

### 2.2. Isolation of the Extensor Digitorum Longus (EDL) Muscle and Primary Myoblast Culture

EDL muscles were obtained from 6-week to 4-month old mice of a C57BL/6 background. The muscles were incubated in collagenase type I (Sigma-Aldrich, C0130, Saint Louis, MO, USA) at 37 °C, 5% CO_2_ for 2 h. The muscles were washed in DMEM, 4.5% glucose (Gibco, 11995, Waltham, MA, USA), containing 1% penicillin/streptomycin (Gibco, 15140, Waltham, MA, USA) and incubated in 5% Bovine serum albumin (BSA)-coated dishes containing DMEM (4.5% glucose, 1% penicillin/streptomycin) for 30 min at 37 °C, 5% CO_2_ to inactivate collagenase. Single muscle fibres were separated by gently blowing with a blunt ended sterilized Pasteur pipette. Subsequently, muscle fibres were seeded in a thin layer matrigel (VWR, 734-0269, Radnor, PA, USA)-coated 6-well plate containing growth medium (DMEM, 4.5% glucose (Gibco, 11995, Waltham, MA, USA), 1% penicillin/streptomycin (Gibco, 15140, Waltham, MA, USA), 10% horse serum (HyClone, 10407223, Marlborough, MA, USA), 30% fetal bovine serum (Biowest, S181B, Nuaillé, France), 2.5ng/mL recombinant human fibroblast growth factor (rhFGF) (Promega, G5071, Madison, WI, USA), and 1% chicken embryonic extract (Seralab, CE-650-J, Huissen, The Netherlands)). Primary myoblasts were allowed to proliferate and migrate off the muscle fibres for 3–4 days at 37 °C, 5% CO_2_. After gentle removal of the muscle fibres, myoblasts were cultured in matrigel-coated flasks until passage 5. Cells were pre-plated in an uncoated flask for 15 min with each passage to reduce the number of fibroblasts in culture. Cell population was 99% Pax7+. All experiments with primary myoblasts were performed on matrigel-coated plates. Primary myoblasts were cultured in differentiation medium for 6 h or allowed to differentiate for 2 days before treatment with 10 ng/mL TGF-β1 (Peprotech, 100-21C, London, UK) or 300 ng/mL myostatin (Peprotech, 120-00, London, UK).

### 2.3. Tgfbr1 and Acvr1b siRNA Assay

C2C12 cells were seeded at a density of 7900 cells/cm^2^ in a 12-well plate (Greiner Bio-One, 665180, Alphen aan den Rijn, The Netherlands) in antibiotic-free growth medium (DMEM, 1% glucose (Gibco, 31885, Waltham, MA, USA), 10% fetal bovine serum (Biowest, S181B, Nuaillé, France)) at 37 °C, 5% CO_2_ and allowed to adhere overnight. SiRNA with a final concentration of 25 nM was prepared according to manufacturer’s protocol. Then, 50 nM siControl, 25 nM siAcvr1b + 25 nM siControl, 25 nM siTgfbr1 + 25 nM or 25 nM siAcvr1b + 25 nM siTgfbr1 was added to the medium of the cells. We used 2 µL DharmaFECT1 per well. The cells were treated with siRNA for 24 h in antibiotic-free growth medium. Subsequently, cells were treated with siRNA for 48 h in antibiotic-free differentiation medium (DMEM, 1% glucose (Gibco, 31885, Waltham, MA, USA), 2% horse serum (HyClone, 10407223, Marlborough, MA, USA)). The following reagents for transfection were obtained from Dharmacon (Lafayette, Colorado): ON-TARGET plus Non-targeting Pool (D-001810-10), DharmaFECT1 (T-2001), 5X siRNA Buffer (B-002000-UB-100), mouse ON-TARGET plus *Tgfbr1* siRNA (J-040617-05), and mouse ON-TARGET plus *Acvr1b* siRNA(J-043507-08)

### 2.4. Ctgf and Fgf-2 siRNA Assay

C2C12 myoblast cells were seeded at a density of 4200 cells/cm^2^ and cultured in antibiotic-free growth medium (DMEM, 4.5% glucose (Gibco, 11995, Waltham, MA, USA), 10% fetal bovine serum (Biowest, S181B, Nuaillé, France)) at 37°C, 5% CO_2_. The cells were transfected with siRNA targeting *Ctgf* or *Fgf-2* (Ambion^®^ Silencer^®^ Select Pre-Design siRNA, Ctgf siRNA ID: s66077, Fgf-2 siRNA ID: s201344, Carlsbad, CA, USA) or a siRNA-negative control (Silencer^®^ Select Negative Control #1 siRNA, Invitrogen 4390843, Carlsbad, CA, USA). SiRNA was re-suspended to a final concentration of 10 µM and lipofectamine transfection reagent (Lipofectamine^®^ RNAiMAX Reagent, Invitrogen 13778100, Carlsbad, CA, USA) was used to prepare the siRNA–lipid complex according to manufacturer’s protocol for a 24-well plate set-up. The cells were cultured for 24 h in antibiotic-free growth medium and transfected with *Ctgf* or *Fgf-2* siRNA–lipid complex for another 24 h. Cells were transfected a second time in antibiotic-free differentiation medium (DMEM, 4.5% glucose, 2% horse serum (HyClone, 10407223, Marlborough, MA, USA). After 16 h, the cells were treated with TGF-β1 (10ng/mL) for 0 h, 3 h and 48 h.

### 2.5. RNA Isolation and Reverse Transcription

Cells were lysed in TRI reagent (Invitrogen, 11312940, Carlsbad, CA, USA). After this, 10% bromochloropropane (Sigma-Aldrich, B9673, Saint Louis, MO, USA) was added. Lysates were inverted and incubated at room temperature for 5 min and centrifuged (4 °C, 12,000 g, 10 min). The RNA containing supernatant was transferred to a new centrifuge tube and washed with 100% ethanol 2:1. RNA was further isolated using the RiboPure RNA purification kit (Thermo Fisher Scientific, AM1924, Waltham, MA, USA). Then, 500 ng RNA and 4 µL SuperScript VILO Mastermix (Invitrogen, 12023679, Carlsbad, CA, USA) were diluted to 20 µL in RNAse free water and reverse transcription was performed in a 2720 thermal cycler (Applied Biosystems, Foster City, CA, USA), using the following program: 10 min 25 °C, 60 min 42 °C, 5 min 85 °C. The cDNA was diluted 10x in RNAse free water. 

### 2.6. Quantitative Real Time PCR

We added 7.5 µL Fast SYBR Green master mix (Fischer Scientific, 10556555, Pittsburgh, PA, USA), 2.5 µL primer mix and 5 µL cDNA in duplo in a 48-well plate. The program ran on the StepOne real time PCR (Applied Biosystems, Foster City, CA, USA) was 20 s at 95 °C holding stage, 40 times 3 s 95 °C step 1 and 30 s 60 °C step 2 cycle stage, 15 s 95 °C, 1 min 60 °C and 15 s 95 °C. *Gapdh* was used as a housekeeping gene to correct for cDNA input. The efficiency of all used primers (Table 1) was tested.

### 2.7. Western Blotting

Cells were lysed in RIPA buffer (Sigma-Aldrich, R0278, Saint Louis, MO, USA) containing 1 tablet of protease inhibitor (Sigma-Aldrich, 11836153001, Saint Louis, MO, USA) and 1 tablet of phosStop (Sigma-Aldrich, 04906837001, Saint Louis, MO, USA) per 10 mL. The total protein concentration in the lysates was determined using a Pierce BCA Protein Assay kit (Thermo Scientific, 23225, Waltham, MA, USA). The absorbance was measured using a microplate spectrophotometer (Epoch Biotek, Winooski, VT, USA) and the protein concentration was calculated using Gen5 software (BioTek, Winooski, VT, USA). An 8% polyacrylamide gel was made. Then, 15 µL sample mix, containing 9 µg total protein and 5 µL sample buffer (5.7 mL water, 1.6 mL glycerol, 1.1 mL 10% SDS, 1.3 mL 0.5 M Tris (pH6.8), 25 mg dithiotreitol (DTT), 300 µL bromophenol blue) was heated to 90 °C for 5 min, cooled on ice and loaded onto the gel. The gel was run in electrophoresis buffer (25 mM Tris base, 190 mM glycine, 0.1% SDS) at 70 V until the samples reached the separating gel and then run at 150 V until the samples reached the bottom of the gel. Next, the proteins were transferred onto a polyvinylidene fluoride (PVDF) membrane (GE Healthcare, 15269894, Chicago, IL, USA) for 1 h at 80V on ice in cold blot buffer (25 mM Tris base, 190 mM glycine, 20% ethanol). The membrane was rinsed in water and washed 2x in Tris-buffered saline and Tween-20 (TBS-T) (20 mM Tris/HCl, 137 mM NaCl, 0.1% Tween-20). The membrane was incubated for 1 h in 2% enhanced chemiluminescence (ECL) prime blocking agent (GE Healthcare, RPN418, Chicago, IL, USA) in TBS-T at 4 °C while shaking. Subsequently, the membrane was incubated overnight in 2% blocking agent in TBS-T with primary antibody (Table 2) at 4 °C while shaking. The membrane was washed 3 × 5 min in TBS-T and incubated in 2% blocking agent in TBS-T with secondary antibody (Table 2) for 1 h at room temperature. ECL solution A and B (GE Healthcare, RPN2235, Chicago, IL, USA) were mixed 1:1 at room temperature and the membrane was incubated for 5 min. Images were taken by the ImageQuant LAS500 (GE healthcare, life sciences, Chicago, IL, USA) and relative intensity of protein bands was quantified using ImageJ [26]. Pan actin was used as a loading control.

### 2.8. Immunofluorescence 

Cells were washed 2x with cold phosphate-buffered saline (PBS) (Gibco, 14190250, Waltham, MA, USA) and fixated for 10 min in 4% paraformaldehyde (PFA) (Fisher Scientific, Pittsburgh, PA, USA) at room temperature. Cells were washed 3x in PBS and permeabilised in 0.1% Triton X-100 in PBS for 10 min. After this, the cells were washed 3x in PBS with 0.05% Tween20 (PBS-T) and incubated for 1 h in 5% normal goat serum (ThermoFisher Scientific, 50062Z, Waltham, MA, USA) in PBS at room temperature. The cells were incubated overnight with primary antibody (Table 2) in 5% normal goat serum in PBS at 4 °C. Then, the cells were washed 3 × 5 min in PBS-T and incubated with secondary antibody (Table 2) in PBS-T for 1 h at room temperature. Cells were washed again 3 × 5 min in PBS-T and incubated in PBS with 4‘,6-diamidino-2-phenylindole (DAPI) (100 ng/mL). After this, the cells were rinsed with PBS and stored in PBS at 4 °C. Images were taken with a fluorescent microscope (Zeiss Axiovert 200M, Hyland Scientific, Stanwood, WA, USA) using the program Slidebook 5.0 (Intelligent Imaging Innovations, Göttingen, Germany). The images were analysed using ImageJ [26]. 

### 2.9. Statistical Analysis

Graphs were made in Prism version 8 (GraphPad software, San Diego, CA, USA). All data were presented as mean + standard error of the mean (SEM). The data were normalised by the values of a control group or of control cells at 0 h. In graphs of time-dependent relative mRNA expression, values of control cells at 0 h were not presented. Statistical analysis was performed in SPSS version 25 (IBM, Amsterdam, The Netherlands). Significance in the difference between two groups was determined by independent t-test. Statistical significance for multiple comparisons was determined by one-way analysis of variance (ANOVA) or two-way ANOVA with post-hoc Bonferroni corrections. Significance was set at * *p <* 0.05. 

## 3. Results

### 3.1. TGF-β Inhibits Expression of Myogenic Genes in both C2C12 Myoblasts and Myotubes

TGF-β reduced both the fusion index (number of myotubes with two or more nuclei per total number of nuclei) and differentiation index (number of nuclei within the myotubes per total number of nuclei) of C2C12 cells (Figure 1a–d). After 1 h of TGF-β treatment, in both myoblasts and myotubes SMAD2 and SMAD3 were phosphorylated (Figure 1e–i), indicating that both myoblasts and myotubes are sensitive to TGF-β. SMAD phosphorylation was inhibited by TGF-β receptor type I inhibitor Ly364947. 

Subsequently, to assess acute and delayed effects in both myoblasts and myotubes, the time-dependent effects of TGF-β on myogenic gene expression were examined after 1, 3, 9, 24 and 48 h of treatment. After 9, 24 and 48 h of TGF-β treatment, *Myod* mRNA expression levels in myoblasts were reduced compared to those in untreated cells, although, after 48 h, *Myod* mRNA expression levels were increased compared to those at earlier time points (Figure 2a). After 24 and 48 h, myogenin (*Myog*) and embryonic myosin heavy chain (*Myh3*) mRNA expression levels in myoblasts were reduced compared to those in untreated cells, although expression levels did gradually increase compared to earlier time points (Figure 2b,d). These results show that TGF-β does not acutely reduce the expression levels of *Myod*, *Myog* and *Myh3* in myoblasts, but rather reduces or attenuates differentiation-related increases in mRNA expression levels of *Myod*, *Myog* and *Myh3* at later time points. In myotubes, *Myog* mRNA expression levels were not significantly affected by TGF-β (Figure 2c). However, after 24 and 48 h of TGF-β treatment, *Myh3* expression levels were reduced compared to those in untreated myotubes (Figure 2e). Thus, TGF-β represses *Myh3* mRNA expression, even in differentiated myotubes.

Regarding the mechanisms underlying effects on differentiation, inhibitor of differentiation 1 (*Id1*) overexpression has been suggested to inhibit differentiation [27]. Since TGF-β induces *Id1* expression in various cell types via SMAD1/5 [28], we quantified *Id1* expression levels. In both C2C12 myoblast and myotubes, TGF-β transiently upregulated *Id1* expression after 1 h (Figure 2f,g), which corresponded with observed SMAD1/5 phosphorylation (Appendix A). In myoblasts, after 24 and 48 h of TGF-β treatment *Id1* mRNA expression levels were slightly reduced compared to those in untreated cells, whereas in myotubes *Id1* mRNA expression levels remained elevated. Based on these results, together with the known function of *Id1*, it is conceivable that *Id1* is involved in TGF-β mediated inhibition of differentiation.

In addition, effects of TGF-β on cell cycle inhibitor cyclin-dependent kinase inhibitor 1A (*Cdk1na*) mRNA expression was examined, because myostatin has been suggested to inhibit myoblast differentiation through inhibition of cyclin-dependent kinase inhibitor 1A [21]. In untreated C2C12 myoblasts, after 48 h *Cdk1na* mRNA expression levels were increased, while this increase was inhibited by TGF-β treatment (Figure 2h). However, *Cdk1na* expression increased during myoblast differentiation (Figure 2i) and no significant effects were observed at earlier time points. This indicates that effects on *Cdkn1a* mRNA expression were likely related to inhibited differentiation, rather than a direct effect of TGF-β on *Cdkn1a* mRNA expression. This suggests that TGF-β does not inhibit differentiation via the regulation of *Cdkn1a* expression.

### 3.2. TGF-β Does Not Affect Myotube Size In Vitro

TGF-β does not only negatively regulate muscle mass via the inhibition of myoblast differentiation, but TGF-β overexpression in adult mouse muscle has also been shown to result in increased expression of E3 ligase atrogin-1, as well as a reduction in muscle fibre cross sectional area [18]. Furthermore, myostatin is well known to stimulate the expression of E3 ligases, both in adult muscle as well as in myotubes in vitro [29]. E3 ligases are involved in protein degradation via Akt/FOXO signalling and play a role in muscle atrophy [30]. These studies indicate that TGF-β may induce protein degradation and subsequent muscle fibre atrophy via a similar mechanism as myostatin does. In addition, TGF-β-induced protein degradation in differentiating myoblasts may attenuate further myoblast differentiation. Since it remains to be assessed whether TGF-β induces muscle atrophy directly via upregulation of E3 ligase expression, time-dependent effects of TGF-β treatment on expression of muscle specific E3 ligases were determined. In myoblasts, after 3, 9, 24 and 48 h of TGF-β treatment mRNA expression levels of muscle RING-finger 1 (*Murf-1*) were reduced, while after 24 h *Atrogin-1* mRNA expression was transiently repressed (Figure 3a,b). In myotubes, the expression levels of *Atrogin-1* were not affected by TGF-β, whereas after 24 and 48 h mRNA expression levels of *Murf-1* were reduced compared to those in untreated myotubes (Figure 3c,d). These results suggest that TGF-β may protect myotubes against E3 ligase-induced protein degradation. However, our results also show that the endogenous expression of *Murf-1* and *Atrogin-1* increased during differentiation, which suggests that the observed effects of TGF-β on *Murf-1* and *Atrogin-1* expression levels were likely related to its inhibitory effect on differentiation (Figure 3e,f). In both myoblasts and myotubes, TGF-β transiently increased the expression levels of the ligase *Musa1* (Figure 3g,h). In myoblasts, *Musa1* expression levels were significantly increased after 9 h. In myotubes, expression levels were increased after 3 and 9 h. 

Subsequently, myotube thickness was measured in C2C12 myoblasts that were differentiated in the presence or absence of TGF-β for three days (cells shown in Figure 1a). There was no significant difference in diameter between myotubes treated with TGF-β and controls (Figure 3n). Furthermore, while SMAD2 and SMAD3 were phosphorylated after 1 h of TGF-β treatment, no significant effects on Akt or ERK1/2 phosphorylation were observed (Figure 3j–m). Together, these results indicate that in vitro TGF-β alone does not affect myotube size.

### 3.3. TGF-β Affects Fibrotic Gene Expression in a Time-Dependent Manner in Both Myoblasts and Myotubes

Time-dependent effects of TGF-β on fibrotic gene expression in C2C12 myoblasts and myotubes were studied. In both myoblasts and myotubes, TGF-β acutely and transiently induced the expression of *Ctgf* and *Fgf-2* (Figure 4a–d). Expression levels peaked between 3 and 9 h of treatment and remained significantly increased for at least 48 h compared to levels in untreated cells. In myoblasts, after 3 h TGF-β treatment, *Col1a1* expression levels were 1.9-fold higher compared to those in untreated cells. This effect gradually increased, and after 48 h, *Col1a1* expression levels were 5.6-fold higher in comparison to levels in untreated cells (Figure 4e). In myotubes, *Col1a1* mRNA expression levels were 10-fold higher compared to those in myoblasts (Figure 4i). After 9 and 48 h of TGF-β treatment, *Col1a1* expression levels were 1.5-fold higher compared to those in untreated cells (Figure 4f). NAPDH oxidase 4 (*Nox4*) is a TGF-β target gene that is required for TGF-β-induced expression of components of extracellular matrix (ECM) [31]. Our results show that in both myoblasts and myotubes, *Nox4* mRNA expression levels were significantly higher compared to those in untreated cells, after 9 or 3 h of TGF-β treatment, respectively. The effect of TGF-β treatment gradually increased and after 48 h, in myoblasts, *Nox4* expression levels were 7.9-fold higher and, in myotubes, 3.1-fold higher compared to those of untreated cells (Figure 4g,h). These results suggest that TGF-β stimulates fibrosis by increasing collagen type I expression in both myoblasts and myotubes.

### 3.4. TGF-β Induces Col1a1 Expression via Ctgf and Fgf-2 in Myoblasts

To investigate whether TGF-β induces *Col1a1* expression in C2C12 myoblasts directly or via the autocrine expression of *Ctgf* or *Fgf-2*, the effects of TGF-β treatment on *Col1a1* expression were studied in the presence of siRNA targeting *Ctgf* or *Fgf-2*. At all time points, treatment with siRNA reduced *Ctgf* or *Fgf-2* mRNA expression levels compared to levels of control siRNA treatment by >90% and >80%, respectively (Figure 4j,m). At 48 h of TGF-β treatment the induction of *Col1a1* mRNA expression was substantially lower (approximately 50%) in the presence of siRNA targeting either *Ctgf* or *Fgf-2* compared to controls (Figure 4k,n), suggesting that *Col1a1* mRNA expression is at least in part regulated by TGF-β dependent *Ctgf* and *Fgf-2* expression. In addition, after 3 h of TGF-β treatment, *Ctgf* knockdown did not affect *Fgf-2* mRNA expression, although after 48 h *Fgf-2* mRNA expression was significantly lower (approximately 70%) in the presence of siRNA targeting *Ctgf*, compared to controls (Figure 4l). *Ctgf* mRNA expression was significantly lower (>55%) in the presence of siRNA against *Fgf-2* compared to controls at all time points (Figure 4o). 

### 3.5. TGF-β Has a Larger Effect on Muscle Differentiation and Fibrosis than Myostatin

Due to the functional and mechanistic similarities between TGF-β and myostatin, the effects of myostatin and TGF-β on C2C12 and primary myoblasts were studied. C2C12 and primary myoblasts, as well as myotubes, were treated with different doses of myostatin or TGF-β. Although a higher concentration of myostatin was needed compared to that of TGF-β in C2C12 and primary myoblasts, as well as myotubes, both proteins induced the translocation of SMAD2 to the nucleus. Figure 5a shows that in primary myotubes and undifferentiated myoblasts, 1 h of 10 ng/mL TGF-β or 300 ng/mL myostatin treatment resulted in the nuclear translocation of SMAD2. Little effect was observed for 0.01 ng/mL TGF-β or 10 ng/mL myostatin. Both of these ligands have a molecular weight of 25 kDa. In primary myoblasts, the comparison of effects of 3 and 48 h myostatin or TGF-β treatment on myogenic and fibrotic gene expression levels showed that after 48 h TGF-β reduced *Myh3* expression by approximately twofold compared to controls, while myostatin did not affect *Myh3* expression (Figure 5b,c). Furthermore, although in primary myoblasts after 3 h of treatment both myostatin and TGF-β significantly enhanced *Ctgf* mRNA expression levels, TGF-β increased *Ctgf* expression levels by 2.2-fold, while myostatin increased *Ctgf* expression levels only by 1.6-fold (Figure 5d,e). These results indicate that TGF-β has a stronger effect on fibrotic and myogenic gene expression levels than myostatin.

### 3.6. Tgfbr1 Levels Correlate with Ctgf and Col1a1 Expression Levels

To further examine the effects of myostatin and TGF-β, type I receptors *Acvr1b* and *Tgfbr1* were individually or simultaneously blocked in myoblasts using specific siRNAs. C2C12 myoblasts were treated with siRNA against *Acvr1b* or *Tgfbr1* for 24 h in growth medium and were additionally treated with siRNA for 48 h in differentiation medium. *Acvr1b* and *Tgfbr1* siRNA reduced receptor mRNA levels by >60% and >50%, respectively, without affecting expression of the other receptor (Figure 5f,g). No significant effects of siRNA treatment on *Myod* or *Myog* expression levels were observed (Figure 5h,i). In line with these results, receptor blocking during differentiation using chemical blocker Ly364947 did not affect fusion or differentiation index nor myotube thickness after 3 days of differentiation (Figure 6a–e). In addition, Ly364947 treatment did not affect *Myh3* expression levels after 48 h of differentiation. However, when C2C12 myoblasts were simultaneously treated with TGF-β and Ly364749, *Myh3* mRNA expression levels were significantly increased compared to those in TGF-β treated cells and similar to those in control cells. In line with observations in primary myoblasts, in C2C12 cells myostatin treatment had no significant effect on *Myh3* expression (Figure 6f). In addition, when the receptors were blocked with Ly364947 during proliferation for 24 h and subsequent differentiation for 2 days, *Myh3* expression was significantly increased (Figure 6g). Knockdown of *Acvr1b* did not significantly affect levels of *Ctgf* and *Col1a1* mRNA, whereas *Tgfbr1* knockdown reduced expression levels of *Ctgf* and *Col1a1* mRNA. Combined knockdown of *Acvr1b* and *Tgfbr1* did not reduce *Ctgf* or *Col1a1* mRNA levels significantly further than *Tgfbr1* knockdown (Figure 5j,k). In addition, *Tgfbr1* mRNA expression levels significantly correlated with both *Ctgf* and *Col1a1* mRNA expression levels (Figure 5l,m). Taken together, these results indicate that TGF-β signalling via *Tgfbr1* has a stronger effect on muscle fibrosis compared to myostatin.

## 4. Discussion

The aim of this study was to assess the time-dependent effects of TGF-β signalling on gene expression in myoblasts and myotubes and compare the effects of TGF-β and myostatin signalling in myoblasts. Here we show that in vitro TGF-β treatment inhibits the expression of a subset of myogenic genes in both myoblasts and myotubes, but does not affect myotube thickness. Most importantly, our results show that TGF-β regulates the expression of fibrotic genes in both myoblasts and myotubes in a time-dependent manner. TGF-β regulates *Col1a1* mRNA expression at least in part via *Ctgf* and *Fgf-2* and, in addition, *Ctgf* and *Fgf-2* are also required to induce the expression of each other. Moreover, our results show a more prominent role for TGF-β in SMAD signalling, as well as myogenic and fibrotic gene expression in comparison to myostatin.

### 4.1. TGF-β Affects Myogenic Gene Expression in Both Myoblasts and Myotubes

TGF-β is known for its inhibitory effect on myoblast differentiation in vitro through inhibition of MyoD [14,32]. As expected, TGF-β inhibited myoblast differentiation and myogenic gene expression. Also, in myotubes, a reduction in *Myh3* expression was observed after 24 and 48 h of TGF-β treatment. Embryonic myosin heavy chain (eMHC), which is encoded by *Myh3*, is normally only expressed during embryonic/fetal and neonatal development, but is transiently re-expressed during muscle regeneration. The loss of eMHC in adult muscle in vivo has been shown to change MHC isoform expression, while in vitro *Myh3* knockdown may result in reduced fusion index and a reduced number of reserve cells, which suggests that loss of *Myh3* results in the early differentiation of MuSCs, depleting the MuSC pool [33]. Together, these results suggest that long-term TGF-β expression in muscle fibres after injury or in chronic disease may impede proper regeneration through repression of *Myh3* expression. 

Additionally, TGF-β has been known from previous studies to interfere with MyoD function via two different mechanisms. First, TGF-β-induced SMAD3 can directly interact with MyoD. Second, TGF-β/SMAD3 interferes with the interaction between MyoD and myocyte enhancer factor 2 (MEF2), which is required for the expression of many myogenic genes [14,32]. Here, we show that TGF-β induces *Id1* expression acutely and transiently in both myoblasts and myotubes. *Id1* is known to inhibit myoblast differentiation by interfering with the formation of MyoD/E complexes, which are required for MyoD function [27]. Our data suggest that the upregulation of *Id1* mRNA may be another mechanism through which TGF-β interferes with MyoD function.

### 4.2. TGF-β Does Not Affect Myotube Size In Vitro

TGF-β overexpression within mouse muscle has been shown to result in the stimulation of atrogin-1 expression and atrophy in vivo [17,18]. To investigate whether this increase in atrogin-1 expression was a direct or indirect effect of TGF-β, time-dependent effects of TGF-β on E3 ligase mRNA expression were studied. In contrast to what has been shown in vivo, C2C12 myotubes did not show evidence for any effect of TGF-β on muscle atrophy. TGF-β treatment resulted in a reduction in *Atrogin-1* and *Murf-1* mRNA expression, rather than an increase. Moreover, an increase in both *Atrogin-1* and *Murf-1* expression was observed during differentiation, which suggests that the observed TGF-β-induced effects on E3 ligase mRNA expression were likely related to inhibition of differentiation. In both myoblasts and myotubes, expression levels of the ligase *Musa1* were transiently increased. Furthermore, TGF-β did not affect Akt or ERK1/2 phosphorylation nor myotube size. Together, these data indicate that in C2C12 myotubes, TGF-β does not directly contribute to atrophy. However, in vivo long term overexpression of TGF-β may lead to a reduction in muscle fibre size [17,18]. Based on our data, this observed in vivo TGF-β overexpression-induced atrophy is possibly mediated via *Musa1* rather than by elevated *Murf-1* or *Atrogin-1* expression levels. Furthermore, we show that TGF-β stimulates *Nox4* and *Id1* mRNA expression. These genes have been implied to play a role in muscle atrophy [34,35]. TGF-β has been shown to induce caspase 3 expression and DNA fragmentation in C2C12 cells [36]. As such, myonuclear apoptosis and loss of muscle stem cells induced by TGF-β may contribute to muscle atrophy as well. The role of TGF-β in the regulation of muscle fibre size requires further investigation.

### 4.3. TGF-β Contributes to Fibrosis by Stimulation of Fibrotic Gene Expression in Myoblasts and Myotubes

Our data show that both myoblasts and myotubes express various pro-fibrotic genes and TGF-β stimulates the expression of these genes in a time-dependent manner. This suggests that, in addition to its effect on fibroblasts, TGF-β likely also contributes to muscle fibrosis through effects on myoblasts and muscle fibres. The stimulatory effects of TGF-β on *Col1a1* mRNA expression in myotubes were relatively small compared to those in myoblasts. Nevertheless, myotubes may contribute substantially to collagen type I production. Basal expression levels of *Col1a1* in myotubes were approximately 10-fold higher than in myoblasts. Moreover, MuSCs comprise approximately 2%–5% of the myonuclei within mature muscle [37] and the number of fibroblasts is roughly 10-fold lower than the number of myonuclei [38,39]. Therefore, it is conceivable that within mature skeletal muscle, differentiating myoblasts and muscle fibres contribute substantially to the production of collagen type I. 

Collagen type I is found in the endo-, peri- and epimysium surrounding muscle fibre [40,41]. Collagen fibres reinforce the ECM surrounding muscle fibres, which is essential in providing a niche for MuSCs, giving structure to the muscle and is even crucial for proper muscle function [42,43,44]. It is conceivable that during myogenesis and muscle regeneration, muscle fibres will secrete collagen type I to contribute to the deposition of connective tissue that provides a scaffold for the regenerating parts of the muscle fibre. However, chronic high expression of TGF-β in skeletal muscle may contribute to muscle fibrosis via the continuous elevated expression of collagen. In muscular dystrophies and aged muscle, TGF-β expression in damaged areas of the muscle may cause excessive collagen deposition. This may result in locally enhanced stiffness along the muscle fibre, which may cause strain distributions along the length of the muscle fibre. As a consequence, muscle fibres are likely to become susceptible to further injuries. In addition, excessive collagen deposition will result in enhanced stiffness of the muscle stem cell niche and likely alter MuSC mechanosensitivity, which may reduce myoblast differentiation and thus impair muscle regeneration capacity [44,45,46,47]. 

Besides pro-fibrotic growth factors and ECM genes, TGF-β also induced the expression of *Nox4*. *Nox4* expression is induced by TGF-β within various cell types such as endothelial cells or lung mesenchymal cells [31,48]. *Nox4* is part of an enzyme family which catalyses the reduction of oxygen into reactive oxygen species (ROS). In lung fibrosis, *Nox4*-dependent H_2_O_2_ generation is required for TGF-β mediated myofibroblast differentiation and ECM production [31]. Furthermore, *Nox4* is a known source for oxidative stress in many tissues and in chronic kidney disease both *Nox4* and oxidative damage markers are increased in muscle [49]. Therefore, we suggest that prolonged TGF-β expression in muscle wasting disorders may contribute to oxidative damage via *Nox4* upregulation. 

### 4.4. TGF-β Induces Col1a1 Expression via Autocrine Ctgf and Fgf-2 Signalling

In lung fibrosis, TGF-β is known to induce collagen 1 expression via CTGF [50,51,52]. This, in combination with the observed expression patterns for *Ctgf*, *Fgf-2* and *Col1a1* in our myoblasts and myotubes, raised the question regarding whether in muscle cells TGF-β directly induced *Col1a1* expression or indirectly via enhancement of expression of these growth factors. *Ctgf* and *Fgf-2* were significantly knocked down using siRNA. After 48 h of TGF-β treatment, *Col1a1* mRNa expression levels were significantly reduced when *Ctgf* or *Fgf-2* was knocked down. This suggests that *Col1a1* expression is at least in part dependent on both *Ctgf* and *Fgf-2* expression in an autocrine manner. In corneal endothelial cells and human vertebral bone marrow stem cells, FGF-2 has been implied to stimulate collagen production [53,54], while in muscle FGF-2 is best known to stimulate MuSC activation and proliferation [55,56]. In this study, we show for the first time that in C2C12 muscle cells *Fgf-2* is required for TGF-β induced *Col1a1* mRNA expression.

Our results show that after 3 h of TGF-β treatment, *Ctgf* knockdown did not significantly affect *Fgf-2* expression; however, *Fgf-2* expression was significantly reduced after 48 h of TGF-β treatment in the presence of siRNA against *Ctgf*. These data suggest that TGF-β acutely induces *Fgf-2* expression independently of changes in *Ctgf* expression, though chronic expression of *Fgf-2* depends on *Ctgf* expression levels. *Ctgf* expression was shown to depend on *Fgf-2* levels both acutely and chronically. To the best of our knowledge, this interaction has not been reported before. See Figure 7 for a schematic of the proposed mechanism for TGF-β induced regulation of *Ctgf*, *Fgf-2* and *Col1a1.* We suggest that TGF-β stimulates *Col1a1* expression largely via the autocrine and paracrine signalling of *Ctgf* and *Fgf-2* and that *Ctgf* and *Fgf-2* may regulate the expression of each other via a positive feedback loop. 

### 4.5. TGF-β Has a More Pronounced Effect than Myostatin on Myoblast Differentiation and Fibrotic Gene Expression

Because of overlap in functional implications and mechanistic similarities between TGF-β and myostatin signalling in myoblasts, we compared the effects of both growth factors on myogenic and fibrotic gene expression. In order to induce downstream activation of SMAD2 signalling, a higher concentration of myostatin was required compared to TGF-β. Furthermore, in myoblasts, TGF-β had a larger effect on *Myh3* and *Ctgf* expression compared to myostatin. To further compare effects of TGF-β and myostatin signalling on myoblasts, these ligands were inhibited by using siRNA against their type I receptors. TGF-β is best known to signal via the TGF-β type I receptor TGFR-1 [24]. In epithelial cells, it has been shown that myostatin can signal via TGFR-1, as well as via ACTR-1B [23]. In mouse myoblasts, myostatin has been shown to signal mainly via ACTR-1B and not via TGFR-1, while in mouse fibroblasts myostatin signals mainly via ACTR-1B [25]. Together, these studies suggest that the knockdown of *Tgfbr1* mainly inhibits TGF-β signalling, while *Acvr1b* knockdown inhibits myostatin signalling. Here, we show in C2C12 myoblasts that *Ctgf* and *Col1a1* mRNA levels correlate with *Tgfbr1* mRNA expression levels, but not with *Acvr1b* expression levels. Moreover, no synergistic effects on the expression of pro-fibrotic genes were observed for combined receptor knockdown. Together, these data indicate that in muscle cells TGF-β has a more pronounced effect on fibrosis than myostatin and that pro-fibrotic gene expression in muscle is mainly mediated via *Tgfbr1*, and not via *Acvr1b*. 

*Acvr1b* and *Tgfbr1* inhibition using siRNA did not affect the expression of myogenic genes. Furthermore, we showed that receptor blocking during differentiation using chemical blocker Ly364947 did not affect the differentiation or fusion index after 3 days, nor the expression of *Myh3* after 2 days. However, in cells treated with both TGF-β and Ly364947, *Myh3* expression levels were similar to those of control cells, which indicates that Ly364947 cancels out the negative effect of TGF-β on *Myh3* expression levels. In addition, when *Acvr1b* and *Tgfbr1* receptors were blocked by Ly364947 during proliferation for 24 h and subsequent differentiation for 2 days, *Myh3* expression levels were significantly increased compared to those of controls. This indicates that the negative effects of TGF-β on myoblast differentiation were cancelled by *Tgfbr1* blocking. The role of *Acvr1b* in myoblast differentiation cannot be concluded based on these results.

Under differentiation conditions, receptor blocking does not further enhance the expression of myogenic genes, which indicates that effects of TGF-β and possibly myostatin on differentiation are dose-dependent and time-dependent. The serum levels (i.e., growth factors such as TGF-β) are relatively low in the differentiation medium compared to the levels in growth medium. This suggests that low concentrations of TGF-β have a minor effect on myogenic gene expression and myoblast differentiation. Note that there is a difference between the chemical blocker Ly364947 and the siRNAs targeting *Acvr1b* and *Tgfbr1* in interference with type I receptor function. While Ly364947 blocks TGF-β signalling within one hour, as demonstrated in Figure 1, siRNAs interfere with the translation of the target mRNA, which may result in a delayed knockdown of type I receptors (Figure 5). This may explain why the presence of siRNA in growth medium did not affect myogenic gene expression. Based on our results, it seems that myoblast differentiation is less sensitive to myostatin signalling than to TGF-β signalling.

### 4.6. Implications in Therapeutic Treatments

Altogether, our results demonstrate that TGF-β signalling has an inhibitory effect on myoblast differentiation and contributes substantially to fibrosis. Therefore, the TGF-β pathway proves to be an interesting potential therapeutic target for treatment of muscular dystrophies. The inhibition of the TGF-β pathway may relieve and attenuate progressive muscle pathology characterized by severe fibrosis and loss of muscle mass. However, taking into account that TGF-β affects various cellular processes throughout the body, generic inhibition of the protein may have serious consequences. Our data show that TGF-β inhibits differentiation and induces fibrosis directly via its receptor in myoblasts and differentiated myotubes. This indicates that the inhibition of TGF-β exclusively within muscle tissue may be an effective approach to improve muscle regeneration in muscular dystrophy. Furthermore, our data demonstrate that TGF-β has a larger effect on differentiation and fibrosis than myostatin. Moreover, *Tgfbr1*, but not *Acvr1b* inhibition, significantly reduced *Ctgf* and *Col1a1* mRNA expression levels, while simultaneous receptor knockdown did not reduce expression levels even further. This suggests that solely blocking *Tgfbr1* and concomitant inhibition of TGF-β signalling may be sufficient to reduce fibrosis in muscular dystrophy. However, when in pathological conditions, both the inhibition of fibrosis and improved regeneration are required, thus simultaneous blocking of the *Tgfbr1* and *Acvr1b* receptor may be desirable. It has been shown that both myostatin and activins signal via *Acvr1b* and that these ligands synergistically inhibit regulation of muscle size [57,58]. Thus, simultaneous targeting of *Tgfbr1* and *Acvr1b* in vivo may still have a synergistic effect on overall muscle function improvement. 

## 5. Conclusions

In conclusion, our data show that TGF-β inhibits myogenic gene expression in both myoblasts and myotubes, but does not affect myotube size in vitro. Most importantly, our results show that TGF-β stimulates *Col1a1* mRNA expression largely via autocrine expression of *Ctgf* and *Fgf-2*. Moreover, the effects of TGF-β on myogenic and fibrotic signalling are more pronounced than those of myostatin. Knockdown of *Tgfbr1* was sufficient to decrease *Ctgf* and *Col1a1* expression levels, while knockdown of *Acvr1b* had little effect. These results indicate that during muscle regeneration, TGF-β induces fibrosis via *Tgfbr1* by stimulating autocrine signalling of *Ctgf* and *Fgf-2.*

## Figures and Tables

**Figure 1 cells-09-00375-f001:**
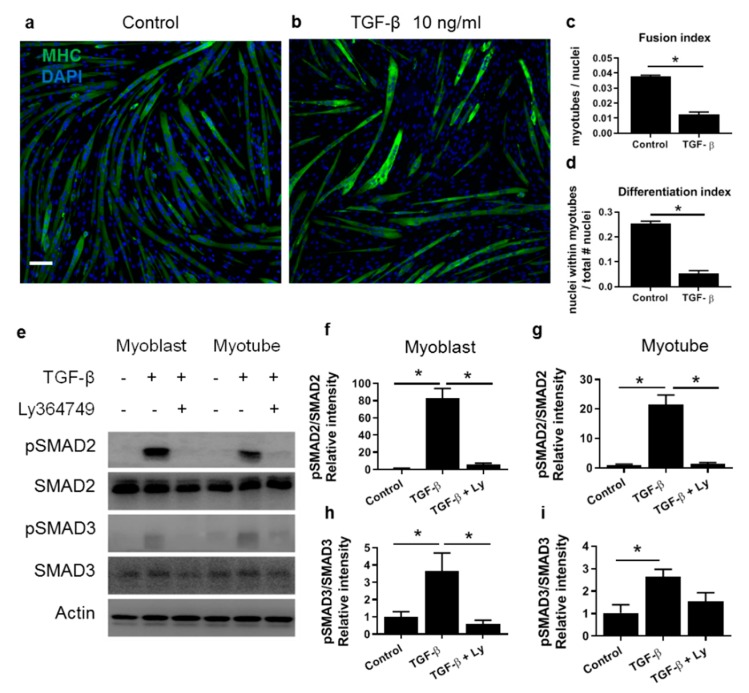
Transforming Growth Factor β (TGF-β) inhibits C2C12 differentiation. (**a**,**b**) C2C12 cells were induced to differentiate in control medium (**a**) or medium supplemented with TGF-β (**b**). Myotubes stained for myosin heavy chain (MHC) (green). Nuclei were stained using DAPI (blue). Scale indicates 100 µm. (**c**,**d**) Fusion index, defined as number of myotubes ≥ 2 nuclei/total number of nuclei and differentiation index defined as number of nuclei within MHC+ myotubes/total number of nuclei were reduced after TGF-β. (**e**,**f**,**g**,**h**,**i**) In both myoblasts and myotubes, phosphorylation levels of SMAD2 (**f**,**g**) and SMAD3 (**h**,**i**) were increased upon 1 h of TGF-β treatment. Pan actin served as loading control. Phosphorylation levels are displayed as relative intensity of pSMAD/total SMAD. Data were normalized to values of control condition. Error bars indicate standard error of the mean; * indicates significant difference at *p <* 0.05; *n =* 4 experiments per condition.

**Figure 2 cells-09-00375-f002:**
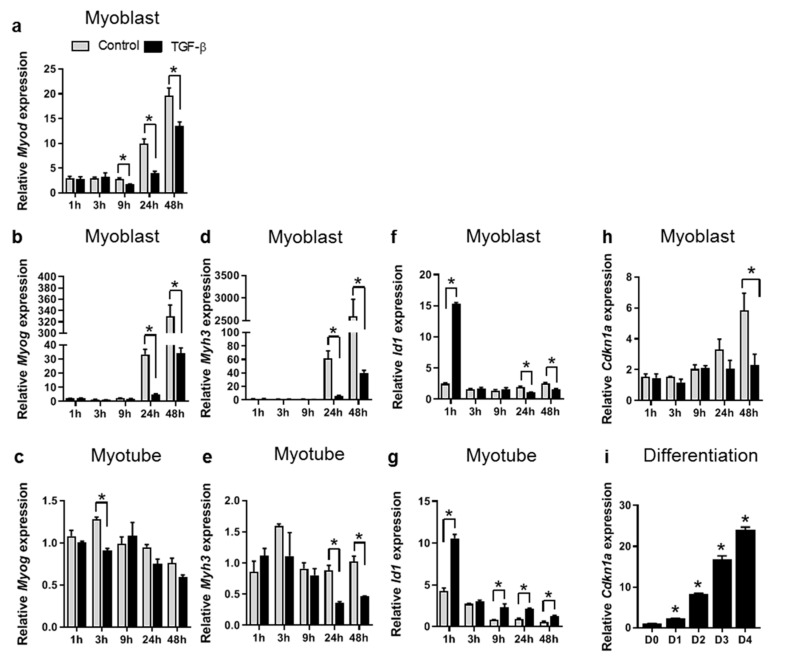
TGF-β reduces myogenic gene expression. (**a**,**b**,**d**,**f**) In myoblasts, expression levels of *Myod* (**a**) were reduced by TGF-β after 9 h compared to those of untreated cells, while expression levels of *Myog* (**b**) and *Myh3* (**d**) were reduced after 24 h. *Id1* expression levels (**f**) were induced after 1 h and repressed after 24 and 48 h. (**c**,**e**,**g**) In myotubes, the expression levels of *Myog* (**c**) were unaffected by TGF-β, while expression levels of *Myh3* (**e**) were reduced compared to those of untreated cells after 24 and 48 h. Expression levels of *Id1* (**g**) were induced after 1 h and remained slightly elevated at later time points. (**h**) In untreated myoblasts, the expression levels of *Cdkn1a* significantly increased, while this increase was inhibited in TGF-β treated cells. (**i**) *Cdkn1a* mRNA expression increased during myoblast differentiation, where D0 is the start of differentiation and D1, 2, 3 and 4 are days 1 to 4 of differentiation. *Gapdh* served as housekeeping gene. Data were normalized to values of control cells at 0 h; * indicates significant difference at *p <* 0.05; *n =* 3 experiments per condition.

**Figure 3 cells-09-00375-f003:**
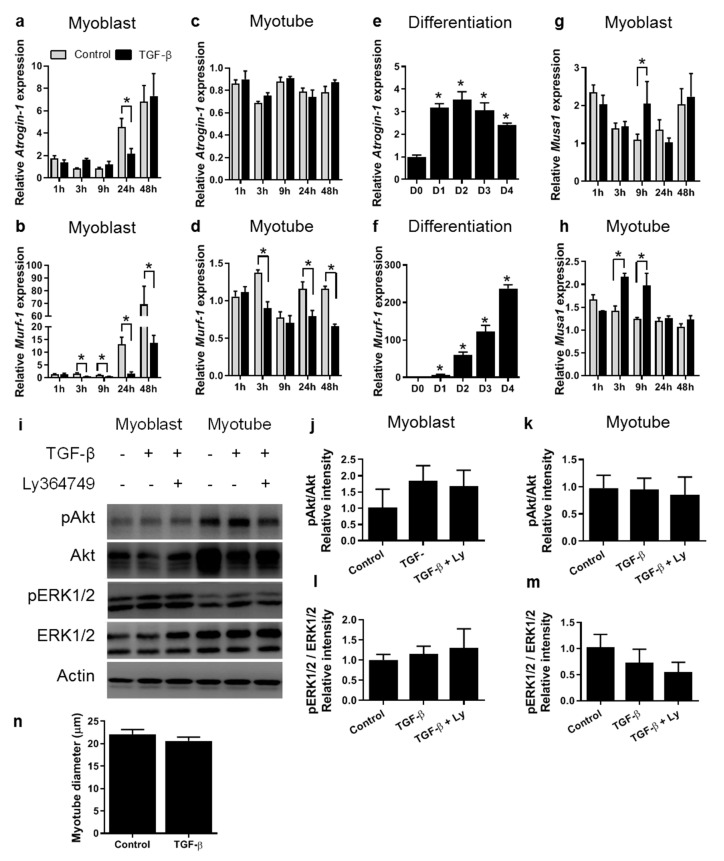
TGF-β does not affect myotube size. (**a**,**b**) In myoblasts, relative expression levels of *Atrogin-1* (**a**) and *Murf-1* (**b**) were inhibited by TGF-β. (**c**,**d**) In myotubes, *Atrogin-1* (**c**) expression was unaffected by TGF-β, while *Murf-1* (**d**) expression levels were inhibited after 3, 24 and 48 h. (**e**,**f**) mRNA expression of *Atrogin-1* (**e**) and *Murf-1* (**f**) increased during differentiation, where D0 is the start of differentiation and D1, 2, 3 and 4 are days 1 to 4 of differentiation. (**g**,**h**) In both myoblasts (**g**) and myotubes (**h**), TGF-β transiently increased *Musa1* expression levels. *Gapdh* served as housekeeping gene. Data were normalized to values of control cells at 0 h; * indicates significant difference at *p <* 0.05; *n* = 3 experiments per condition (**a**–**h**). (**i**,**j**) Western blot quantification of Akt phosphorylation in myoblasts, (**k**) Akt phosphorylation in myotubes, (**l**) ERK1/2 phosphorylation in myoblasts, (**m**) ERK1/2 phosphorylation in myotubes. Pan actin served as loading control. Phosphorylation levels are displayed as relative intensity of phospho/total protein. Data were normalized to values of the control condition. Error bars indicate standard error of the mean; *n =* 4 experiments per condition. **n** After 3 days of TGF-β treatment, the myotube diameters displayed in Figure 1a were not significantly different from those of control condition. *n =* 80 per experimental condition.

**Figure 4 cells-09-00375-f004:**
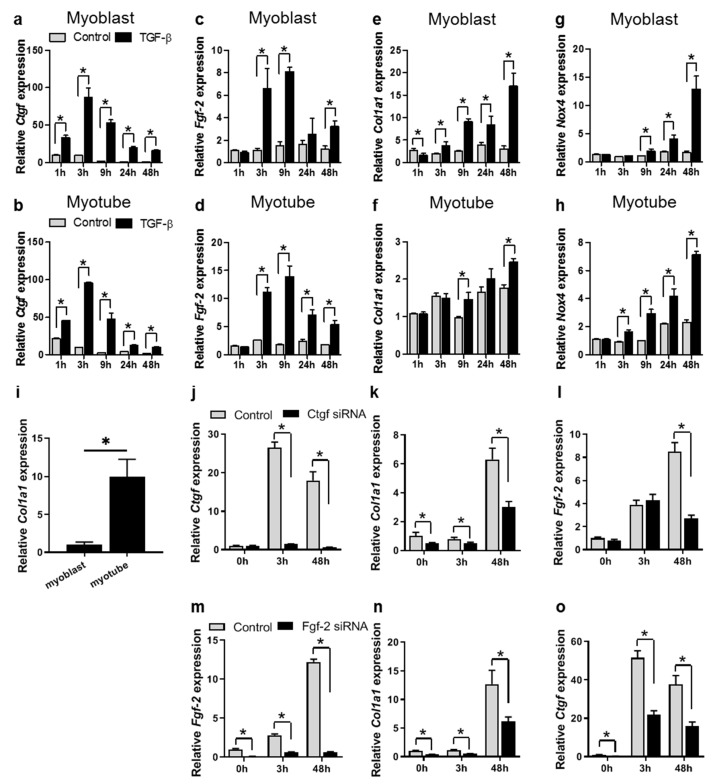
TGF-β affects fibrotic gene expression levels in myoblasts and myotubes in a time-dependent matter. (**a**–**h**) mRNA expression levels of (**a**) *Ctgf* in myoblasts, (**b**) *Ctgf* in myotubes, (**c**) *Fgf-2* in myoblasts, (**d**) *Fgf-2* in myotubes, (**e**) *Col1a1* in myoblasts, (**f**) *Col1a1* in myotubes, (**g**) *Nox4* in myoblasts, (**h**) *Nox4* in myotubes. (**a–d**) In myoblasts and myotubes, expression levels of *Ctgf* and *Fgf-2* were acutely induced by TGF-β. (**e–h**) *Col1a1* and *Nox4* expression levels were gradually induced by TGF-β. *Gapdh* served as housekeeping gene; data were normalized to values of control cells at 0 h; * indicates significant difference at *p* < 0.05; *n* = 3 experiments per condition (**a**–**h**). (**i**) *Col1a1* mRNA expression levels in myotubes were approximately 10-fold higher compared to those in myoblasts. *Gapdh* served as housekeeping gene; data were normalized to expression values in myoblasts. * *p* indicates significant difference at <0.05; *n* = 6 experiments per condition. (**j**) At all time points after siRNA treatment, *Ctgf* expression was knocked down by >90%. (**k**) *Col1a1* expression was reduced in the presence of siRNA targeting *Ctgf*. (**l**) After 3 h of TGF-β treatment, *Fgf-2* expression increased independent of *Ctgf*. After 48 h of TGF-β treatment, in the presence of siRNA targeting *Ctgf, Fgf-2* expression was significantly reduced. **m** At all time points after siRNA treatment, *Fgf-2* expression was knocked down by >81%. Both *Col1a1* (**n**) and *Ctgf* (**o**) expression levels were significantly reduced in the presence of siRNA targeting *Fgf-2* compared to those of control siRNA condition. *Gapdh* served as housekeeping gene; data were normalized to values of control cells at 0 h; * indicates significant difference at *p <* 0.05; *n =* 6 experiments per condition (**j–o**).

**Figure 5 cells-09-00375-f005:**
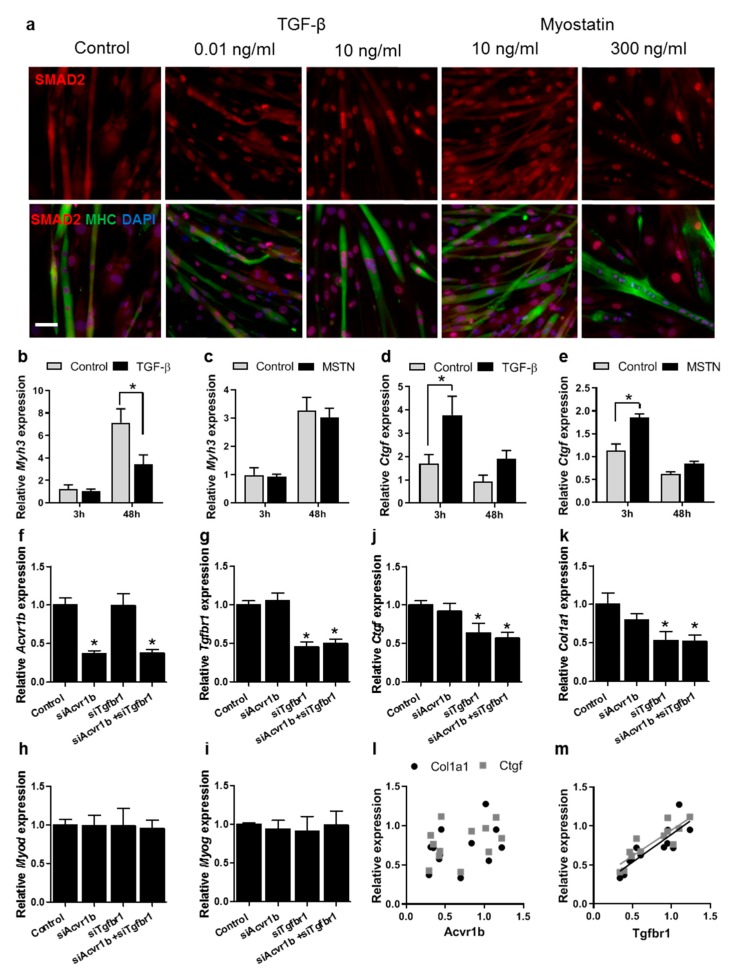
TGF-β has a larger effect on myoblasts compared to myostatin. (**a**) Primary cells were induced to differentiate for 2 days and subsequently treated with TGF-β (0.01 or 10 ng/mL) or myostatin (10 or 300 ng/mL). Myotubes were stained for MHC (green) and nuclei were stained using DAPI (blue). SMAD2 is visible in red. The scale indicates 100 µm. After TGF-β or myostatin treatment, SMAD2 was translocated to the nucleus in both myotubes and undifferentiated myoblasts compared to controls. (**b**,**c**) In primary mouse myoblasts, expression levels of *Myh3* mRNA were reduced after 48 h TGF-β treatment compared to those in untreated cells (**b**), while no differences were observed for myostatin (MSTN) (**c**). (**d**,**e**) Expression levels of *Ctgf* were increased after 3 h TGF-β (**d**) or myostatin treatment (**e**) compared to those in untreated cells, although TGF-β had a larger effect. (**f**,**g**) Treatment with specific siRNAs reduced levels of *Acvr1b* (**f**) or *Tgfbr1* (**g**). (**h**,**i**) Knockdown of *Acvr1b* or *Tgfbr1* did not affect *Myod* (**h**) or *Myog* (**i**) expression levels. (**j**,**k**) *Tgfbr1* knockdown slightly reduced *Ctgf* (**j**) and *Col1a1* (**k**) expression levels, while *Acvr1b* knockdown had little effect. The combined knockdown of *Tgfbr1* and *Acvr1b* did significantly reduce *Ctgf* and *Col1a1* expression levels. *Gapdh* served as housekeeping gene; error bars indicate standard error of the mean; * indicates significant difference at *p <* 0,05; *n =* 3 experiments per condition; data were normalized to values of control cells at 0 h. (**l**,**m**) There is a significant correlation between *Tgfbr1* expression level and *Ctgf* and *Col1a1* expression levels (**l**), while no such correlations were found between *Acvr1b* expression levels and *Ctgf* or *Col1a1* expression levels (**m**).

**Figure 6 cells-09-00375-f006:**
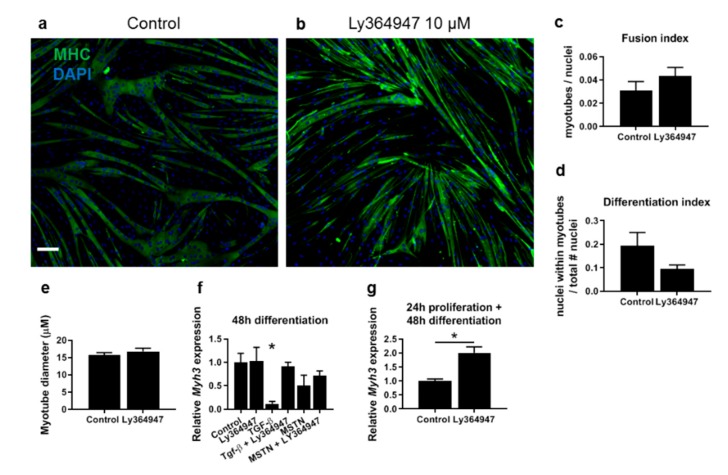
Effects of type I receptor blocking on myoblast differentiation is time-dependent. (**a**,**b**) C2C12 cells were induced to differentiate in control medium (**a**) or in medium supplemented with the TGF-β receptor inhibitor Ly364947 (**b**). Myotubes were stained for MHC (green) and nuclei were stained using DAPI (blue). Scale indicates 100 µm. (**c**) Fusion index, defined as number of myotubes ≥ 2 nuclei/total number of nuclei and (**d**) differentiation index defined as number of nuclei within MHC+ myotubes/total number of nuclei were not significantly affected by Ly364947 treatment. Error bars indicate standard error of the mean; * indicates significant difference at *p <* 0.05; *n* = 4 experiments per condition. (**e**) Myotube thickness was not affected by Ly36447 treatment, compared to control condition. (**f**) 48 h of Ly364947 treatment did not affect *Myh3* expression levels in differentiating C2C12 myoblasts. *Myh3* expression levels were significantly increased in cells simultaneously treated with TGF-β and Ly364947 compared to those of cells treated with TGF-β and similar to those of untreated cells. Myostatin did not significantly affect *Myh3* expression levels. (**g**) *Myh3* expression levels were significantly increased when C2C12 myoblasts were treated with Ly364947 during proliferation for 24 h and subsequent culture in differentiation medium for 48 h.

**Figure 7 cells-09-00375-f007:**
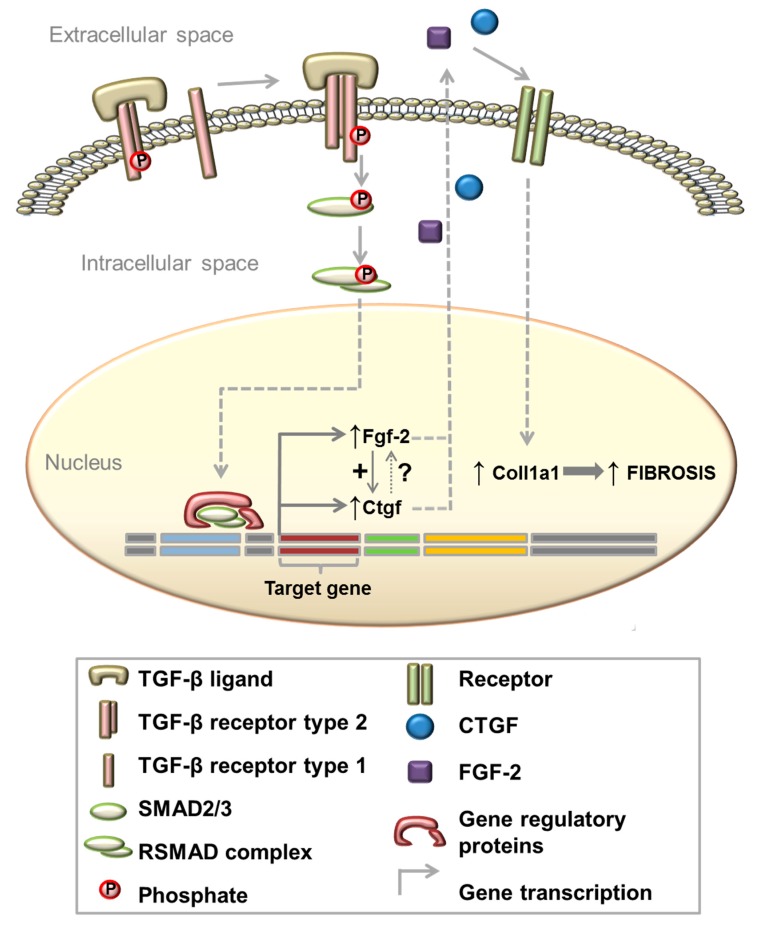
Schematic illustration of a proposed mechanism of how TGF-β regulates *Col1a1* mRNA expression. TGF-β binds to its receptors and activates downstream SMAD2/3 signalling. Subsequently, R-SMAD complexes translocate into the nucleus to regulate mRNA expression of growth factors such as *Ctgf* and *Fgf-2*. CTGF and FGF-2 proteins are secreted by the muscle cell and subsequently induce *Col1a1* expression via autocrine or paracrine signalling. Furthermore, expression levels of *Fgf-2* and *Ctgf* are dependent on each other.

**Table 1 cells-09-00375-t001:** Primers for qPCR.

Primer	Sequence
mGapdh-forward	TCCATGACAACTTTGGCATTG
mGapdh-reverse	TCACGCCACAGCTTTCCA
Myod1-forward	AGCACTACAGTGGCGACTCA
Myod1-reverse	GCTCCACTATGCTGGACAGG
Myog-forward	CCCAACCCAGGAGATCATTT
Myog-reverse	GTCTGGGAAGGCAACAGACA
Myh3-forward	CGCAGAATCGCAAGTCAATA
Myh3-reverse	CAGGAGGTCTTGCTCACTCC
Ctgf-forward	CCACCCGAGTTACCAATGAC
Ctgf-reverse	GCTTGGCGATTTTAGGTGTC
Fgf-2-forward	AAGCGGCTCTACTGCAAGAA
Fgf-2-reverse	GTAACACACTTAGAAGCCAGCAG
Col1a1-forward	ATGTTCAGCTTTGTGGACCT
Col1a1-reverse	CAGCTGACTTCAGGGATGT
Id1-forward	ACCCTGAACGGCGAGATCA
Id1-reverse	TCGTCGGCTGGAACACAT
Nox4-forward	CTTTTCATTGGGCGTCCTC
Nox4-reverse	GGGTCCACAGCAGAAAACTC

**Table 2 cells-09-00375-t002:** Antibodies for Western Blotting and immunofluorescence.

AB	Dilution	Experiment	Company
Phospho-SMAD2 (Ser465/467) Rabbit mAb	1:1000	WB	cell signaling/3108, Leiden, The Netherlands
SMAD2 Rabbit mAb	1:1000, 1:200	WB, IF	cell signaling/5339
Phospho-SMAD3 (S423/425) Rabbit mAb	1:1000	WB	cell signaling/9520
SMAD3 Rabbit mAb	1:1000	WB	cell signaling/9523
Phospho-Akt (Ser473) Rabbit mAb	1:2000	WB	cell signaling/4060
Akt (pan) Rabbit mAb	1:1000	WB	cell signaling/4691
Phospho-ERK1/2 Rabbit mAb	1:2000	WB	cell signaling/4370
ERK1/2 Rabbit mAb	1:4000	WB	cell signaling/4695
Pan actin Rabbit mAb	1:1000	WB	cell signaling/8456
Myosin, sarcomere (MHC)	2.5 µg/mL	IF	DSHB/MF20-s, Iowa City, IA, USA
Anti-Rabbit IgG-POD (LumiLightPLUS Western Blotting Kit)	1:2000	WB	Roche/12015218001, Basel, Switzerland
Goat anti-Rabbit IgG (H + L), Alexa Fluor^®^ 555 conjugate	1:500	IF	ThermoFisher Scientific/A21428, Waltham, MA, USA
Goat anti-Mouse IgG (H + L), Alexa Fluor^®^ 488 conjugate	1:250	IF	ThermoFisher Scientific/A11001

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
