# Peer review of "TGF-β Regulates Collagen Type I Expression in Myoblasts and Myotubes via Transient Ctgf and Fgf-2 Expression"

_cells, 2020, doi:10.3390/cells9020375_

Round 1

Reviewer 1 Report

In this manuscript by Hillege et al. the authors examine how TGFb signaling can affect collagen expression in cultured myoblasts and myotubes. They use both genetic as well as pharmacological approaches to dissect the signaling changes that cause upregulation of Col1a1 levels. While the story presented is of interest, some modifications would help increase the interest;

They state in the title that TGFb induces collagen expression, however they only show one specific collagen isoform. They should show if also other collagens are regulated this way to make such a general statement in the title Figure 2 states that there is an inhibition of myogenic signaling after TGF blockade, however, it is only reduced, not blocked. It is not clear what added value there is in the analysis of the E3 ligases in figure 3. The data shown in figure 1 using the inhibitor are somewhat in contrast to what is shown in figure 5H and 5I. The inhibitor strongly affects MyoD and myogenin expression levels in myoblasts, however genetic inhibition of its main targets does not do anything. It would be useful to combine the genetic approach with the inhibitor to see if its inhibitory effect in myoblasts is actually going through TGFRB1

Reviewer 2 Report

This is a very well written manuscript with data supporting most conclusions.  I would like to thank the authors for such comprehensive description of materials and methods which is highly commendable. Specific comments:

Myotubes are differentiated myoblasts and the authors suggest that TGFb may inhibit myotube differentiation, perhaps this needs to be reconsidered; Atrogin-1 ang MuRF-1 expression would be more relevant to myotube atrophy, whereas in myoblasts it would be amore appropriate to investigate markers of apoptosis; Increased collagen synthesis was noted in both myoblasts and myotubes following TGFb treatment – can the authors discuss the physiological consequences of this, especially in mytoubes? Would this change the properties of myofibres or rather have an effect on the niche? Did the cells change phenotype following TGFb treatment? Whilst changes in expression of fibrosis-related genes were shown following TGFb treatment, fibrosis itself has not been demonstrated on a phenotypic level; Myotube diameter should be measured for all experiments, e.g. Fig.6.

Round 2

Reviewer 1 Report

The revised version is much improved and acceptable for publication in its current form